# Hardness of Resin Cements Polymerized through Glass-Ceramic Veneers

**DOI:** 10.3390/dj9080092

**Published:** 2021-08-09

**Authors:** Hanan Aldryhim, Omar El-Mowafy, Peter McDermott, Anuradha Prakki

**Affiliations:** 1Faculty of Dentistry, Princess Nourah Bint Abdulrahman University, Riyadh 11564, Saudi Arabia; 2Faculty of Dentistry, University of Toronto, Toronto, ON M5G 1G6, Canada; oel.mowafy@utoronto.ca (O.E.-M.); peter.mcdermott@utoronto.ca (P.M.); anuradha.prakki@dentistry.utoronto.ca (A.P.)

**Keywords:** resin cement, veneers, glass ceramic, hardness

## Abstract

(1) Background: The aim of this study is to evaluate the hardness of resin cements polymerized through ceramic disks under different process factors (ceramic type and thickness, light-polymerization units and polymerization time); (2) Method: Three types of ceramic blocks were used (IPS e.max CAD; Celtra Duo; VITABLOCS). Ceramic disks measuring 0.5 mm, 1.0 mm and 1.5 mm were cut from commercial blocks. Two resin cements (Rely X Veneer and Variolink Esthetic) were polymerized through the ceramic specimens using distinct light-polymerization units (Deep-cure; Blue-phase) and time intervals (10 and 20 s). Hardness of cement specimens was measured using microhardness tester with a Knoop indenter. Data were statistically analyzed using factorial ANOVA (α = 5%); (3) Results: Mean microhardness of Rely X Veneer cement was significantly higher than that of Variolink Esthetic. Deep-cure resulted in higher mean microhardness values compared to Blue-phase at 0.5- and 1-mm specimen thicknesses. Moreover, a direct correlation was found between polymerization time and hardness of resin cement; (4) Conclusions: Surface hardness was affected by resin cement type and ceramic thickness, and not affected by ceramic types, within evaluated conditions. Increasing light-polymerization time significantly increased the hardness of the cement.

## 1. Introduction

Porcelain laminate veneers are increasingly attractive option for restoring anterior teeth mainly because of their esthetic appearance, durability and biocompatibility [1]. Feldspathic porcelains are the most common type of ceramics used for porcelain laminate veneers because of their great esthetic value. They are highly translucent and imitate natural dentition [2]. Recently, interest has been growing in the use of lithium disilicate glass ceramics for porcelain laminate veneers because of their optical properties and superior mechanical properties compared to existing glass ceramics [3].

Light-polymerized resin cements are preferred in veneer cementation because of their extended working time and color stability [4]. Layton reported the clinical success rate of veneering restoration using resin cement as 96% at five to six years, 93% at 10 to 11 years and 91% at 12 to 13 years [5]. Another study reported a success rate of 98.8 percent after six years [6]. The success of resin cements require a clear understanding of the factors that affect its clinical performance [7]. Polymerization is an important factor affecting the properties of resin cements [8,9]. The degree to which resin cements are polymerized is proportional to the amount of light to which they are exposed [10]. However, as light passes through the bulk of the restorative material, its intensity is greatly decreased, thus, decreasing the potential of adequate polymerization [11]. Considering that adequate polymerization is crucial in determining the life of resin bonded ceramic restorations, it is necessary to optimize polymerization to enhance the mechanical properties of resin cements [12]. High-intensity LED polymerization units are considered to achieve adequate resin cement polymerization within a short time. This is explained by the radiant energy theory, which indicates that increasing irradiance by using a high-intensity polymerization light results in more photons available per unit time for absorption [13]. Therefore, more photoinitiator molecules react with amines, and more free radicals are available for polymerization [14]. Hardness test is an effective method to asset the polymerization of resin cement [15]. Many studies have shown that hardness has a positive correlation with the degree of conversion of resin cement [16,17,18]. Studies reported that the hardness of resin cements polymerized through ceramic depends on various factors, such as the type of resin cement, ceramic type and thickness, in addition to the polymerization unit and polymerization time [12,19,20].

Therefore, the objective of this study is to evaluate the hardness of two light-polymerized resin cements, in combination with different ceramic types and ceramic thicknesses, as well as polymerization units and polymerization time. The first null hypothesis to be tested is that the ceramic type and ceramic thickness will not affect the hardness values of light-polymerized resin cements. The second null hypothesis to be tested is that the polymerization unit and polarization time will not affect the hardness values of light-polymerized resin cements.

## 2. Materials and Methods

### 2.1. Experimental Design

This study intended to test the hardness of two light-polymerized resin cements (Rely X Veneer and Variolink Esthetic). For each resin cement, the factors under study were (i) ceramic type at three levels (IPS e.max CAD, Celtra Duo and VITABLOCS Esthetic Line), (ii) ceramic thickness at three levels (0.5 mm, 1.0 mm and 1.5 mm), (iii) light-polymerization unit at two levels (Deep-cure (3M/ESPE, St. Paul, MN, USA)\430–480 nm and Blue-phase (Ivoclar-Vivadent, Liechtenstein, Germany)\430–490 nm), and (iv) polymerization time at two levels (10 and 20 s). The association among factors (3 × 3 × 2 × 2) resulted in 36 test groups for each resin cement. The quantitative response variable was the microhardness of resin cements polymerized through ceramic interposition, measured by a Knoop indenter (*n* = 6). Resin cements polymerized without ceramic interposition were used as a control.

### 2.2. Materials

The materials used in this study are outlined in Table 1.

### 2.3. Preparation of Ceramic Sample

Square specimens measuring 16 mm × 16 mm and thickness of 0.5 mm, 1.0 mm and 1.5 mm (thickness tolerance ± 0.1 mm) were cut from respective ceramic blocks (*n* = 3) using a low-speed micro slicing saw with a diamond blade under water cooling (Isomet 1000, Buehler Ltd., Lake Bluff, IL, USA). For standardization purposes, same ceramic disk was used for testing cement microhardness within each evaluated factor. Crystallization and glazing of ceramic disks were completed according to the respective manufacturer’s instructions.

### 2.4. Preparation of Resin Cement Specimens

The resin cements were placed in a metallic ring (10 mm diameter × 0.5 mm thickness) and covered with a Mylar strip. A 50 g weight was gently applied over the cement to standardize resin cement thickness, and the excess cement was carefully removed with a spatula. The ceramic disks were then placed on top of the resin cements + Mylar strip, and the resin cements were polymerized through ceramic disks with the LED light-polymerization units placed 5 mm above the ceramic disk. The resin cement samples were stored in the dark for 24 h at 37 °C.

### 2.5. Microhardness Test

The microhardness of the resin cement specimens was measured at the top surface at the center of the speciments using a Tukon 300 microhardness tester (Acco Industries Inc., Wilson Instrument Division, Bridgeport, CT, USA) with a Knoop indenter and a 50 g weight for 5 s. The indentation was assessed with the aid of the built-in stereomicroscope (20×). The longer diagonal was marked, and the software calculated the Knoop hardness number (KHN) of each indentation. Microhardness values were determined by the average of six random indentations obtained for each group.

### 2.6. Light Attenuation Test Using MARC Patient Simulator

The MARC (Managing Accurate Resin Curing, BlueLight Analytics Inc., Halifax, NS, Canada) patient simulator is a laboratory energy measurement system integrated into a phantom head. It allows for checking and training of the use of light-polymerization units under near-clinical conditions with light sensor of 4 mm in diameter located between the two maxillary central incisors. Light irradiance delivered from the polymerization units were recorded digitally and analyzed by means of a special software package (MARC 3.0.4.0 software) [21].

Light attenuation was determined through the prepared ceramic disks using Deep-cure (3M/ESPE) and Blue-phase (Ivoclar-Vivadent) LED polymerization units. Ceramic disks were seated centric on the anterior light sensor. A 10 s light irradiance delivered from the polymerization units through the ceramic specimen was recorded digitally. For each ceramic specimen, six independent measurements were performed and averaged. Moreover, light irradiance delivered from both Deep-cure and Blue-phase polymerization units without ceramic disk interpositions were recorded as a control.

### 2.7. Statistical Analysis

Data were evaluated by univariate analyses performed using one-way ANOVA and multivariate analysis using factorial ANOVA. Separate models were used for each of the outcomes. Level of statistical significance was set at 5%. All analyses were performed using IBM SPSS software version 24.

## 3. Results

The hardness of Rely X Veneer cement was significantly higher than that of Variolink Esthetic polymerized through all ceramics disk types and thickness (*p* < 0.001) (Figure 1).

Ceramic types showed no statistically significant difference between each other in mean microhardness values; however, they were significantly lower than the mean microhardness of the control group where the cements were polymerized without ceramic disk interposition (*p* < 0.001). Control group for both Rely X Veneer and Variolink Esthetic had the highest hardness. While the hardness of resin cement under 0.5 mm ceramic thickness is significantly higher than 1 mm and 1.5 mm for all ceramic types. However, the difference between 1 mm and 1.5 mm is not statistically significant (*p* = 0.12) (Figure 1).

Moreover, mean hardness of the resin cements when Deep-cure was used were significantly higher than when Blue-phase was used particularly when the ceramic thickness was less than 1.5 mm (*p* < 0.001) (Figure 2). Further, a direct correlation was found between the polymerization time and the microhardness of the cement when the ceramic thickness was 1 mm or more, and a statistically significant interaction effect was detected (*p* < 0.001) (Figure 3).

A statistically significant decrease in light irradiance was found with increasing ceramic thickness (*p* < 0.001) (Table 2). A significant interaction was found between ceramic type and ceramic thickness suggesting that different ceramic types had higher light attenuation at different ceramic thickness (*p* < 0.001). Specifically, IPS e.max CAD had significantly higher light irradiance at 0.5 mm compared to the other types. However, at 1 mm and 1.5 mm Celtra Duo had significantly higher light attenuation compared to the other two ceramic types (Figure 4). The light irradiance of Deep-cure was significantly higher than that of Blue-phase (*p* = 0.001) through all ceramic types and thicknesses (Table 2). 

## 4. Discussion

Resin cement polymerization under ceramic restoration is one of the most important factors that determines the success of porcelain veneer [22]. Inadequate polymerization can result in lower mechanical properties and faster degradation of cement at finish line by the oral fluids and ultimately causing de-bonding [23]. Studies have reported that sufficient light is fundamental for achieving high conversion of light-polymerized resin cements [8]. The amount of light transmitted through ceramic restorations depends on irradiance of the light-polymerization unit and the type, thickness and translucency of the ceramic material [24].

In this study, the first null hypothesis was partially rejected as the ceramic type did not significantly affect hardness values. However hardness values were significantly affected by ceramic thickness. The second null hypothesis was rejected because the polymerization unit and polymerization time affected significantly the hardness of resin cements.

Ceramic types in this study had no significant effect on the microhardness of light-polymerized resin cements. However, ceramics thickness significantly affected the microhardness. Previous studies reported that resin cement polymerized under ceramic disks received lower light irradiance as the thicknesses of ceramic increased, which affect its polymerization and hardness [8,24,25]. Variable mean microharness values of different resin type reported in the present study are likely due to the different compositions of resin cements, including variations in types, loading, and filler size, which all affect the polymerization of the cements [26].

Previous studies have shown a direct correlation between light intensity and hardness of resin cements [15,27]. In this study, hardness of resin cements using Deep-cure was significantly higher than Blue-phase if the ceramic thickness is less than 1.5 mm (*p* < 0.001), and this is due to the light power of Deep-cure being higher than that of Blue-phase. However, at 1.5 mm ceramic thickness the degree of attenuation had a stronger effect than the different in the light power.

The duration of irradiation is another important factor affecting the polymerization of resin cements through ceramic materials [28]. This is explained by radiant exposure (J/cm^2^); that is, the product of irradiance (mW/cm^2^) and light-polymerization time (s) [18], which indicates that an increase in light-polymerization time could be used to compensate for a reduction in irradiance [29]. Light-polymerized resin cement utilizes photoinitiators that are activated exclusively through a certain range of wavelength from light polymerization units [30]. Camphorquinone is the most prolific photoinitiator, though there are varieties of cement that could contain different ones [20]. Clinicians should, therefore, remain aware of the photoinitiator that the resin cement contains, largely because the wavelength of some polymerization units may be unable to match the cement photoinitiator spectrum of absorption. In this study, the wavelength of the LED light-polymerization unit used agrees with the absorption wavelength range of the photoinitiator recommended by the manufacturer of the resin cement (400–500 nm). The present results demonstrated a direct correlation between the polymerization time and the hardness of the cement when the ceramic thickness is 1 mm or more. Moreover, previous studies have reported that different combinations of irradiance values and light-polymerization periods that resulted in similar radiant energy levels produced similar material properties including surface hardness [27]. In this study, light attenuation increased with increasing ceramic disk thickness, which is consistent with what was reported in previous studies [8,25,31].

Ceramic type was a determining factor for light attenuation in this study. A significant interaction was found between ceramic type and ceramic thickness suggesting that different ceramic types have higher light attenuation at different ceramic thicknesses (*p* < 0.001). Specifically, IPS e.max CAD had significantly higher light irradiance (lower light attenuation) at 0.5 mm thickness compared to the other ceramic types. However, at 1 mm and 1.5 mm thickness Celtra Duo had significantly higher light attenuation compared to other two ceramic types. A plausible explanation for the differences in light irradiance transmittance by ceramic type is the role of various crystalline structures of ceramics, including alternate compositions of filler and matrix [32]. IPS e.max CAD attenuate less light as it contains small crystals that are approximately 1.5 μm in size [7]. Celtra Duo contain lithium silicate, which comprise many small, randomly oriented, interlocking plate-like crystals that scatter more light compared to glass ceramic that is without these interlocking crystals. This phenomenon explains the similarity between IPS e.max CAD and Celtra Duo reduced light irradiance through the different thicknesses [20]. The significant light irradiance difference between IPS e.max CAD and Celtra Duo is related to the addition of 8–10% zirconium oxide to the Celtra Duo [33]. VITABLOCS Esthetic Line is a feldspathic porcelain that contains less crystal and more glass and, therefore, light attenuation by the crystal component is relatively low [34]. Given the fact that the ceramic type can cause a significant difference in light transmittance, this difference did not affect the microhardness of resin cement polymerized underneath the different type of ceramic used in this study. 

In this study, the light irradiance of Deep-cure without ceramic interposition was 1470 Mw/cm^2^ and 1200 mW/cm^2^ for Blue-phase, which is consistent with the information reported by the manufacturers. Moreover, the light irradiance of Deep-cure was found to be significantly higher than that of Blue-phase (*p* = 0.001) through all ceramic types and thicknesses which means that the ceramics light attenuation using the Deep-cure was lower than the attenuation achieved with the Blue-phase polymerization unit. The results of this study are in agreement with those of a previous study, which also reported that higher intensity polymerization unit negatively affects the ceramics light attenuation [14,34].

There is a clear need for additional studies to investigate the effect of increasing the polymerization time more than 20 s on the hardness of light-polymerized resin cement. Future studies must compare the behavior of veneers bonding with different ceramic type in oral environments. Long-term clinical studies will eventually produce clear evidence for determining differing clinical performance and protocols for different ceramic types.

## 5. Conclusions

Within the limitations of this study, it is concluded that:
Hardness of the light-polymerized resin cement through ceramics depended on the type of cement, the thickness of the ceramic and polymerization time.Mean microhardness of Rely X Veneer cement was significantly higher than that of Variolink Esthetic cement. Although, a direct correlation was found between polymerization time and mean microhardness for both resin cements.Although light attenuation was significantly affected by the type of glass ceramic, the hardness of resin cement under different type of glass ceramic disks was not significantly difference.

## Figures and Tables

**Figure 1 dentistry-09-00092-f001:**
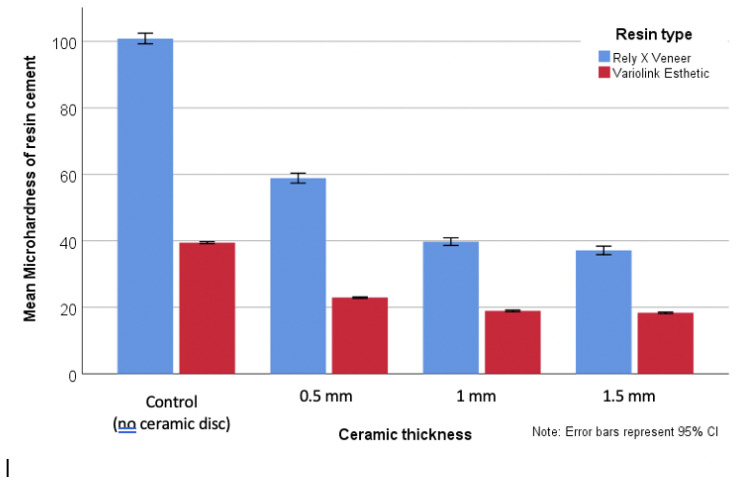
Microhardness of resin cements under ceramic disks.

**Figure 2 dentistry-09-00092-f002:**
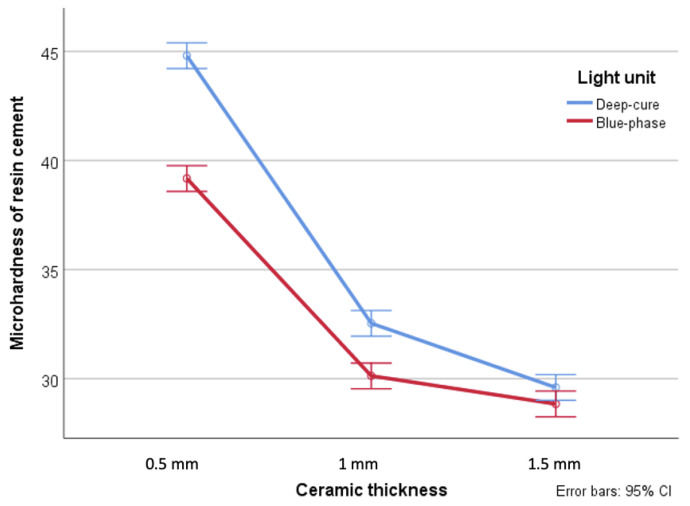
Interaction between ceramic thickness and light polymerization unit.

**Figure 3 dentistry-09-00092-f003:**
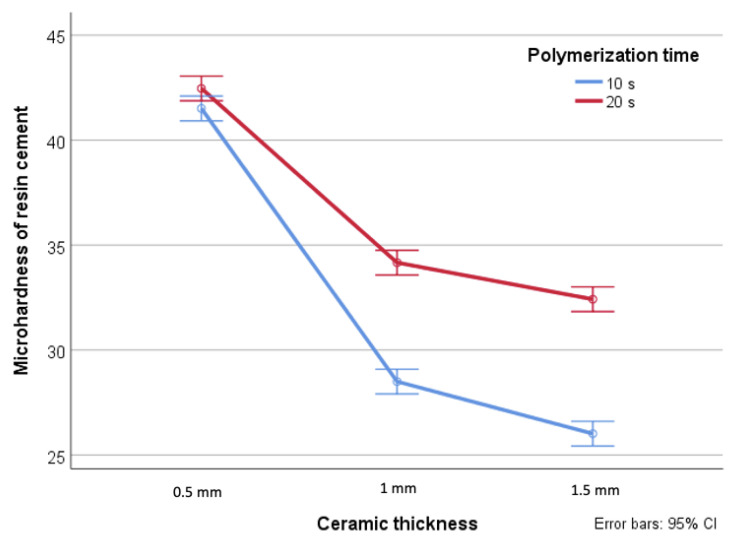
Interaction between ceramic thickness and Polymerization time.

**Figure 4 dentistry-09-00092-f004:**
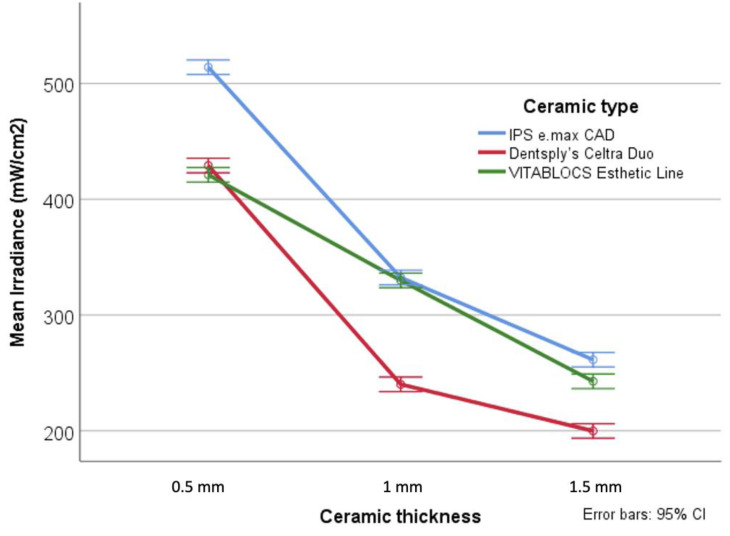
Light irradiance through ceramic disk specimens.

**Table 1 dentistry-09-00092-t001:** Materials used in the study.

Material\Shade	Product Description	Manufacturer	Batch Number
IPS e.max CAD\A2	Lithium disilicate glass-ceramic	Ivoclar Vivadent AG FL-9494Schaan/Liechtenstein Germany	X09583
Celtra Duo\A2	Zircona-reinforced lithium silicate ceramic	DeguDent GmbH Rodenbacher Chaussee 4 63457Hanau-Wolfgang, Germbany	16000013
VITABLOCSEsthetic Line\1M2C	Feldspathic ceramic	VITA Zahnfabrik Spitalgasse3 79713Bad Sackingen, Germany	57090
Rely X Veneer\ Light yellow	Light-polymerize d resin cements	3M ESPE 2510 Conway Avenue St. Paul, MN 55144-1000USA	N952985
Variolink Esthetic\ Neutral		Ivoclar Vivadent AG, Fl-9494 Schaan/Liechtenstein, Germany	X16532

**Table 2 dentistry-09-00092-t002:** The irradiance (mW/cm^2^) of polymerization units through ceramic specimens; different superscript letters indicate statistically significant difference (α = 5%).

Ceramic Type and Thickness		Deep-Cure	Blue-Phase
**IPS e.max CAD**	0.5 mm	566.00 ± 20.97 ^C^	462.00 ± 23.54 ^D^
	1 mm	361.75 ± 9.59 ^F^	303.00 ± 21.73 ^G^
	1.5 mm	286.25 ± 8.11 ^G,H^	236.50 ± 22.06 ^I^
**Celtra Duo**	0.5 mm	473.00 ± 55.17 ^D^	385.21 ± 26.41 ^E,F^
	1 mm	251.75 ± 12.63 ^H,I^	228.50 ± 6.81 ^K^
	1.5 mm	222.54 ± 4.15 ^K^	177.25 ± 9.01 ^L^
**VITABLOCS Esthetic Line**	0.5 mm	450.75 ± 7.12 ^D^	391.42 ± 6.86 ^E^
	1 mm	359.25 ± 9.03 ^F^	300.50 ± 21.52 ^G^
	1.5 mm	270.25 ± 30.47 ^H^	215.25 ± 21.79 ^K^
**Control (no ceramic disk interposition)**		1470 ± 3.25 ^A^	1200 ± 2.7 ^B^

## Data Availability

The data presented in this study are openly available in FigShare at https://doi.org/10.6084/m9.figshare.15105480.v1, reference number (15105480.v1).

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
