# Peer review of "Hardness of Resin Cements Polymerized through Glass-Ceramic Veneers"

_dentistry, 2021, doi:10.3390/dj9080092_

Round 1

Reviewer 1 Report

Minor text editing corrections:

Space should be between the word and the bracket throughout the text, e.g. paragraph 26. biocompatibility[1] correct biocompatibility [1].

In paragraph 81. correct 37°C

In paragraph 263 use the abbreviation of the journal as it is in paragraph 272 Int J Prosthodont

In paragraph 274 use the abbreviation of the journal as it is in paragraph 288 Dent Mater

Author Response

Thank you very much for taking the time to review our manuscript. We appreciate all the valuable comments.

Space was added between the word and the bracket throughout the text.
The temperature corrected to 37°C ( line 91).
Journal abbreviations were corrected through the references.

Reviewer 2 Report

You can find attached the PDF file.

Author Response

Thank you very much for taking the time to review our manuscript. We appreciate all the valuable comments.

The aim of the study was corrected and highlighted in the abstract.

In the introduction, we elaborate more about the hardness test, add more references, and highlight the changes (lines 38-48).

In the material section, we renumbered the subtitles.

We added thickness tolerance for ceramic specimens (line 78).

The distance between the LED light-polymerization units and the samples was added (line 90).

The LED light-polymerization units wavelength was added (lines 65-66).

We mentioned that the used LED light-polymerization units are recommended for the resins tested (lines 215-216).

Temperature corrected to “37°C” (line 91).

For the microhardness test, the Dwell time was added (line 97).

All the indentations were in the center of the specimen's top surface (lines 94-95).

We used the polymerization unit for 10 seconds to record the light attenuation (line113); however, hardness was tested after polymerization for 10 and 20 seconds (line 67).

The LED light-polymerization units wavelength were discussed and highlight (lines 209-217).

Reviewer 3 Report

The paper could be of interest for the journal , the methodology is adequate. The major criticisms is the lack of novelty and the significance of content of the research.  A major revision of the text is required.

Page 1 Remove the number after the keywords and before the abstract divisions.

Introduction:

In the introduction some clincial information should be provided to understand the problematics of polymerization processess.

Authors should discuss some clinical studies where resin based recontruction are followed-up in the long term.

In table 1 please add also the expiring date of tested materials

The discussioni is repetitive. Please explain the meaning of positive correlation and do not repeat the concept (Page 6 and 7).

In the discussion, to better gain the importance of polymerization processess in dentistry, the authors should report and comment some polymerization protocols currently used with the tested cements in restorative dentistry, in post endo reconstruction and in the application of provisional fixed prostheses (such as Maryland bridge).

Reference format should be standardized.

Author Response

Thank you very much for taking the time to review our manuscript. We appreciate all the valuable comments.

We elaborate more regarding some clinical information to explain the problematics of polymerization processes (lines 38-48 and 179-182).

Some clinical studies were added to show the long-term followed-up of veneers restoration using resin cement.

In the discussion, we mentioned the variable two times (in relation to the hardness of the cement and in relation to light attenuation of the ceramic specimens).

In the introduction and in the discussion, we evaporated more to clarified some points and make it more clinical oriented. Both resin cement used in this study were light polymerized resin cement. In the introduction, we add the effect of light intensity on the photoinitiator molecules and polymerization (lines 37-47). In addition, we reemphasized in the discussion part on the importance of choosing the polymerization unit that appropriates for the resin cement used (line 209-217).

Finally, the references were standardized.

Round 2

Reviewer 3 Report

the authors improved the manuscript. Some minor corrections still are needed in english text. 

The paper can be accepted for pubblication